# Toxicity of Synthetic Cannabinoids in K2/Spice: A Systematic Review

**DOI:** 10.3390/brainsci13070990

**Published:** 2023-06-24

**Authors:** Mariana Campello de Oliveira, Mariana Capelo Vides, Dângela Layne Silva Lassi, Julio Torales, Antonio Ventriglio, Henrique Silva Bombana, Vilma Leyton, Cintia de Azevedo-Marques Périco, André Brooking Negrão, André Malbergier, João Maurício Castaldelli-Maia

**Affiliations:** 1Interdisciplinary Group of Alcohol and Drug Studies (GREA), Institute Perdizes, Department of Psychiatry Medical School, São Paulo University, São Paulo 05403-903, SP, Brazilandre.negrao@hc.fm.usp.br (A.B.N.);; 2Department of Psychological Medicine, School of Medical Sciences, National University of Asuncion, San Lorenzo 111421, Paraguay; 3Department of Experimental Medicine, Medical School, University of Foggia, 71122 Foggia, Italy; 4Department of Legal Medicine, Medical School, São Paulo University, São Paulo 05508-090, SP, Brazil; 5Department of Neuroscience, Medical School, FMABC University Center, Santo André 09060-870, SP, Brazil; 6Department of Epidemiology, Mailman School of Public Health, Columbia University, New York, NY 10032, USA

**Keywords:** synthetic cannabinoid, Spice, K2, JWH, AM, RCS, HU, AB-PINACA, APICA, MAM, CHMICA, UR, XLR, AKB, AB-FUBINACA, ADB, EAM, AKB, WIN, PB, MDMB

## Abstract

(1) Background: Synthetic cannabinoids (SCs) are emerging drugs of abuse sold as ‘K2’, ‘K9’ or ‘Spice’. Evidence shows that using SCs products leads to greater health risks than cannabis. They have been associated with greater toxicity and higher addiction potential unrelated to the primary psychoactive component of marijuana, Δ9-tetrahydrocannabinol (Δ9-THC). Moreover, early cases of intoxication and death related to SCs highlight the inherent danger that may accompany the use of these substances. However, there is limited knowledge of the toxicology of Spice ingredients. This systematic review intends to analyze the toxicity of SCs compounds in Spice/K2 drugs. (2) Methods: Studies analyzing synthetic cannabinoid toxicity and dependence were included in the present review. We searched the PubMed database of the US National Library of Medicine, Google Scholar, CompTox Chemicals, and Web of Science up to May 2022. (3) Results: Sixty-four articles reporting the effects of synthetic cannabinoids in humans were included in our review. Ten original papers and fifty-four case studies were also included. Fourteen studies reported death associated with synthetic cannabinoid use, with AB-CHMINACA and MDMB-CHMICA being the main reported SCs. Tachycardia and seizures were the most common toxicity symptoms. The prevalence of neuropsychiatric symptoms was higher in third-generation SCs. (4) Conclusion: SCs may exhibit higher toxicity than THC and longer-lasting effects. Their use may be harmful, especially in people with epilepsy and schizophrenia, because of the increased risk of the precipitation of psychiatric and neurologic disorders. Compared to other drugs, SCs have a higher potential to trigger a convulsive crisis, a decline in consciousness, and hemodynamic changes. Therefore, it is crucial to clarify their potential harms and increase the availability of toxicology data in both clinical and research settings.

## 1. Introduction

Synthetic cannabinoids (SCs) are emerging drugs of abuse sold as ‘K2’, ‘K9’ or ‘Spice’ detected in herbal smoking mixtures. They are one of the drugs labeled as new psychoactive substances (NPS), which are not entirely controlled by the United Nations drugs conventions [1,2]. They have been associated with greater toxicity and higher addiction potential not related to the primary psychoactive component of marijuana, Δ9-tetrahydrocannabinol (Δ9-THC) [3,4,5]. In addition, the first cases of addiction and death related to SCs highlight the danger of their use [6,7,8].

These chemical substances were developed by scientists in the 1970s to study the cannabinoid system and explore its potentially new therapeutic uses, such as the treatment of nausea and pain conditions [6,9]. Chemist John W. Huffman from Clemson University (USA) developed the JWH series, which has a more efficient bond with CB1 receptors [10]. Respectively, compounds JWH-018, JWH-122, and JWH-210 are 5, 60, and 82 times stronger than receiver Δ9-tetrahydrocannabinol (Δ9-THC) [10,11]. SCs have greater toxicity due to the binding power of SCs in cannabinoid receptors, increasing the chance of side effects.

In the early 2000s, substances (e.g., JWH-018) which were primarily created for research purposes started appearing in smoking mixes, and these concoctions for smoking quickly gained popularity, particularly in nations where cannabis use for recreational purposes was prohibited or where users wanted to avoid being detected through standard drug testing. For this reason, several SCs began to be created in secret labs, combined with dried herbal mixes, and sold online as acceptable substitutes for cannabis (or ‘legal highs’). Known as ‘K2’ (in North America), ‘Spice’ (in Europe), ‘Youcatan’, ‘Chill’, or ‘Black Mamba’ and reportedly safe for eating, these concoctions have been widely marketed as smokable herbal combinations [11].

Since then, SCs use has grown considerably, which raises concerns about potential harm. Currently, there are hundreds of SCs known, and the SCs market is constantly evolving, so new compounds are continually being developed. Although often compared to THC, synthetic cannabinoids are not structurally related to the natural cannabinoids present in marijuana. Based on earlier SCs structures, four substructures with an indole, indazole, or carbazol core encircled by various N-substituents were used to create SCs. They form a heterogeneous group, but most SCs are lipid-soluble, nonpolar, composed of 22 to 26 carbon atoms, and very volatile when heated [6].

Nonetheless, anecdotal accounts reported SCs as having deadly side effects, as well as causing neurological disorders (e.g., psychosis, agitation, irritability, paranoia, confusion, anxiety), psychotomimetic effects (e.g., hallucinations, delusions, self-harm), cardiac arrhythmias, several other physical conditions (e.g., tachypnea, hypertension, nausea, vomiting, acute kidney injury, fever, hyperglycemia, hypokalemia, sedation), and fatalities [7,8,12,13,14,15]. Later SCs generations, with psychoactive effects and anecdotal user reports, were released onto the drug market over time, producing severe adverse effects such as increased heart rate, panic attacks, and convulsions [16].

SCs are classified into four distinct generations [17]. The first generation was the pioneer in terms of production; they are CB1 and CB2 full agonists, but with higher affinity than THC and more potent dopamine-stimulating action [18,19]. These include JWH-018, CP-47,497, and HU-210 derivatives [20]. The second generation includes AM-2201 and others from the JWH series, such as JWH-210. An even higher CB receptor affinity is characteristic of the second generation, which distinguishes itself from the first generation only in terms of chemical structure [18]. It corresponds to alkyl derivatives N-methyl piperidine and benzoyl indoles [20]. Compared to the first and second generations, the third generation has the highest affinity to the CB1 receptor, and examples include AB-CHMINACA and MDMB-CHMICA [19]. It presents an indole ring that has been replaced with an indazole or benzimidazole group. This generation also includes compounds with the carbonyl group replaced by a carboxylic or carboxamide group or quinolones with a secondary crystalline structure and nitrogen-containing groups [20]. Lastly, the fourth-generation SCs present an indole or indazole core; an ester, amide, or ketone linker; quinolinyl, naphthyl, adamantyl, tetramethylcyclopropyl, or other moiety ring; and a hydrophobic side chain attached to the nitrogen atom of the indole or indazole core. These main substances are 4F-MDMB-BINACA and 4F-ABINACA [17]. There are no clinical records about toxicological effects or information available as to the pharmacokinetics of the fourth generation.

Despite uncertainty over these substances’ toxicological profiles, the use of SCs is increasing worldwide [21]. Although there are several case reports and observational studies describing various effects (toxicities) of different SCs, to our knowledge, no articles compile those substances’ toxicological effects associated with each SCs. This systematic review’s goal is to summarize and discuss the toxicological effects of SCs.

## 2. Materials and Methods

### 2.1. Eligibility

Original clinical trials analyzing synthetic cannabinoid toxicity and dependence were selected. The following were excluded: non-original studies (i.e., reviews, meta-analysis, discussion articles, study protocols), studies in languages other than English, animal studies not including synthetic cannabinoids abusers or dependents, and studies not analyzing outcomes regarding toxicity and addiction potential.

### 2.2. Information Sources

We searched the PubMed database of the US National Library of Medicine, Google Scholar, CompTox Chemicals, and Web of Science up to May 2022 to identify relevant studies.

### 2.3. Search Strategy

The keywords used in the search were ‘synthetic cannabinoid’ or ‘JWH*’, ‘AM*’, ‘RCS*’, ‘APINACA*’, ‘HU*’, ‘AM*’, ‘AB PINACA*’, ‘APICA*’, ‘MAM*’, ‘CHMICA*’, ‘CP*’, ‘UR*’, ‘XLR*’, ‘AKB*’, ‘AB-FUBINACA*’, ‘ADB*’, ‘EAM*’, ‘AKB*’, ‘WIN*’, ‘PB*’, or ‘MDMB*’. ‘Clinical trial’, ‘Case Reports’, ‘Clinical Study’, ‘Comparative Study’, ‘Evaluation Study’, ‘Multicenter Study’, ‘Observational Study’, ‘Randomized Controlled Trial’, and ‘Human’ (study) were the filters used in PubMed. The saturation strategy was used in the Google Scholar search (i.e., ending the search after three consecutive pages without studies within the topic of the review). The Preferred Reporting Items for Systematic Reviews and Meta-Analyses (PRISMA) were followed for data-gathering purposes. Research methods were registered on OSF https://osf.io/embk8/ (accessed on 15 March 2023).

### 2.4. Study Selection

To select articles for this review, the first and the last authors read the abstracts of all studies found in the search. Duplicate articles were excluded. In the last step, the first author read the remaining studies. Inclusion and exclusion criteria were applied. The present review followed the PRISMA statement for transparent reporting of systematic reviews and meta-analyses [22], as presented in Figure 1.

### 2.5. Data Items

This review searched for information on the following variables: title; author(s); year of publication; country; sample size; mean age; male/female ratio; study setting; study design; type of cannabinoid; dosage; toxic effects; and mortality outcome.

## 3. Results

Sixty-four articles reporting the effects of synthetic cannabinoids in humans were included in our review. Two descriptive analyses of the studies were conducted separately: clinical studies (CS) and case reports (CR). Ten clinical studies and fifty-four case reports were included. Amongst the CS, 359 patients were assessed, with an average of 74% male and 26% female (Table 1). Amongst the CR, data from 234 patients were analyzed, with 30 females (12.8%) and 233 (88.2%) males, with an average age of 26 years (Table 2). Most subjects were recreational users, and 34% had criteria for cannabis use disorder according to the 5th edition of the *Diagnostic and Statistical Manual of Mental Disorders* (DSM-5) [23].

Twenty-six different cannabinoids were identified, including AB-CHMINACA, ADB-CHMINACA, AB-PINACA, ADB-FUBINACA, ADB-PINACA, AB-FUBINACA, MDMB-FUBINACA, MDMB-CHMICA, 5F-ADB, FUB-AMB, 5F-AMB, JWH-018, JWH-073, JWH-122, JWH-022, AM-2201, AM-694, MMB-2201, 5F-PB-22, 5F-AKB-48, PB-22, 6-APB, EAM-2201, BB-22, XLR-11, and UR-144. The most frequent were AB-CHMINACA, ADB-FUBINACA, and JWH-018, respectively. Analyses of SCs in blood, serum, or urine samples were performed using electrospray ionization liquid chromatography-tandem mass spectrometry.

The reported toxicologic effects were seizures, Glasgow Coma Scale (GCS) less than 10, disorientation, psychomotor agitation, red eyes, nausea, vomiting, anxiety, paranoia, palpitations, delirium, psychosis, reckless driving, mood swings, hypothermia, muscular stiffness, nystagmus, dilated pupils, drowsiness, mental confusion, supraventricular tachycardia, respiratory failure, hypoxemia, disseminated vascular coagulation, ischemic stroke, hemiparesis, dysarthria, aphasia, catatonia, persistent hallucination, and ischemic heart disease.

In our review, the most frequent toxic effects mentioned in CS were tachycardia, followed by seizures (Figure 2). In total, 26.1% of CR detailed fatal intoxications from using SCs, particularly AB-CHMINACA, in which 50% of studies detailed the substance’s lethal effects. The reported studies presented severe degrees of intoxication and neuropsychiatric symptoms. In our study, the SCs AB-CHMINACA and ADB-FUBINACA had the highest frequency of symptoms and accounted for 41% of all reported fatalities.

Concerning the CS, four out of ten were conducted in the USA, three in the Netherlands, and one each in Germany, Japan, and the United Kingdom. Six out of ten studies reported using AB-CHIMINACA, which, jointly with MDMB-CHMICA, was associated with the worst outcomes. Nine out of ten studies took place in emergency departments. There was just one randomized controlled study, which was conducted in the Netherlands. Two CS reported death associated with SCs use, with AB-CHMINACA and MDMB-CHMICA being the reported SCs. The prevalence of neuropsychiatric symptoms was higher in third-generation SCs.

Most case reports studies were conducted in the USA, followed by Germany and the United Kingdom. In total, 26.8% (nineteen papers) of the articles related to fatal intoxication. Forty articles (56.3%) had emergency departments (ED) as their setting, 22.5% (sixteen papers) were outside hospital environments, 7% in psychiatric units (five papers), 4% in Institutes of Legal Medicine (three papers), and 10% did not specify (seven papers). Fifty-four studies specified the detected synthetic cannabinoid via either blood or urine clinical testing. Sixteen articles (22.5%) reported concomitant use of other drugs, primarily alcohol and cocaine.

AB-CHIMINACA and ADB-FUBINACA were the two cannabinoids most associated with death outcomes in the included studies, followed by MDMB-CHMICA and 5F-ADB, respectively. Thromboembolic events were reported with ADB-FUBINACA, JWH-018, and XLR-11 (e.g., ischemic heart disease, stroke). XLR-11 also had more association with psychosis, hallucinations, and paranoia. 

The remarkable SCs toxicology findings included the medium concentration in blood for 5F-PB-22 (0.37 ng/mL), AB-CHMINACA (8.2 ng/mL), and 5F-ADB (0.38 ng/mL); in femoral blood for 5F-PB-22 (1.5 ng/mL), XLR-11 (11 ng/mL), ADB-FUBINACA (7.3 ng/mL), MDMB-CHMICA (3.5 ng/mL), EAM-2201 (12.6 ng/mL), PB-22 (1.1 ng/mL), JWH-210 (12 ng/mL), UR-144 (12.3 ng/mL), JWH-022 (3 ng/mL), MDMB-CHMICA (1.7 ng/mL), MDMB-PINACA (4 ng/mL), and FUB-AMB (3.7 ng/mL); in urine for JWH-018 (200 nM); and in postmortem blood levels for ADB-FUBINACA (56 ng/mL). Plasmatic concentrations linked to fatal defects range from 1.4 to 105 ng/mL. In autopsy cases, medium concentration in femoral blood was 0.00009 µg/g for AM-694, 0.0003 µg/g for AM-2201, and 0.00005 µg/g for JWH-018.

## 4. Discussion

SCs are categorized as NPS and have toxic effects which have not yet been fully discussed in the literature. These substances mimic the psychedelic effects of the main phytocannabinoids, the delta-9-tetrahydrocannabinol (THC) [88] and are linked to higher degrees of toxicity in smaller doses. The 64 studies included in our systematic review showed a wide range of dosages of serious chemicals or urine tested in relation to toxic effects, adding to the enormous difficulty in homogenizing effects. They exhibit a variety of toxic side effects which have been described in the literature, such as convulsions, disorientation, forgetfulness, psychomotor agitation, nausea, vomiting, paranoia, palpitations, tachycardia, mental confusion, hypertension, etc. [12,14,89].

SCs toxicity has increased with the affinity and specificity of cannabinoid receptors CB1 and CB2, a phenomenon that may explain their neuropsychiatric effects. The endocannabinoid system’s role in regulating excitability is still not fully understood. Cannabis compounds may have antiepileptic properties but also seem to cause convulsions. Depending on the dose that affects the cerebral regions, endocannabinoids may either stimulate or inhibit the release of GABA, possibly through dopaminergic modulation [90].

The neurotoxicity of these components is limited due to the lack of objective data on SCs. Some studies also reported longer-lasting toxic effects in SCs from third generations and potentially fourth generations due to their connection to increasingly powerful CB1 receptors [17], exhibiting more significant toxicity than earlier generations [20,91,92].

In recent years, the public health field has grown more concerned about the NPS, which include SCs and other psychoactive substances, due to the deaths caused by their abuse. As a result, the number of substances available that can lead to fatal or severe intoxications is increasing yearly. In addition, the NPS metabolic pathways and elimination routes are poorly understood, making it challenging to detect and confirm these substances in toxicological case analyses [93]. There is little detail on how SCs’ metabolic degradation occurs. It is challenging to accurately estimate the lethal SCs concentration in postmortem studies due to the substance’s natural degradation in the bloodstream [25].

NPS draw attention to cases of intoxication in emergency departments when delays in diagnosis can be lethal. These substances are consumed more frequently every day, especially in countries where recreational use is prohibited or where users attempt to evade detection in toxicology tests. Current research indicates that due to the increased intensity of psychological effects and increased likelihood of intoxication, SCs expose healthy patients and individuals with epilepsy to higher risks of intoxication or severe adverse effects [29]. Besides the higher affinity for CB1 receptors, the lack of cannabidiol (CBD) in these substances imposes an increased risk of psychoses and schizophrenia [94,95,96].

The toxic effects of SCs can be compared to those of other NPS. Data on the toxicity of other NPS are still scarce in the literature. Among these are cathinones, also known as ’bath salts‘, which are chemically altered synthetic psychostimulants, similar to the cathinones found in the leaves of the khat plant (*Catha edulis*) [97]. Its toxic effects include clinical complications and severe hemodynamic changes, such as disseminated intravascular coagulation, hyperthermia, arrhythmias, and sialorrhea [98], all of which have been also linked to SCs and may be difficult to differentiate in emergency situations.

Synthetic cannabinoids, interestingly, have a toxicology profile that is quite similar to other synthetic drugs, such as opioids, which have postmortem effects similar to those of SCs. In recent years, the growing number of synthetic opioids, such as fentanyl analogs, has been a source of concern due to their extreme potency, toxicity, and fatality, even at low dosages, as stated by UNODC [1]. Autopsy examinations demonstrated that pulmonary congestion, cardiomegaly, and cerebral edema associated with isotonitazene, followed by metonitazene, were also toxic effects of synthetic opioids [99]. Nonetheless, the toxicity profile observed in SCs differs from other substances directly linked to fatal toxicology cases, such as phenethylamine or phenyl cyclohexyl piperidine (PCP), phencyclidine analogs, and benzodiazepine derivatives. Phenethylamines are well-known amphetamines that include compounds such as 2C, NBOMe, NBOH, benzofurans (for example, Bromo-Dragonfly), and others (6-APB, PMMA) [100,101].

Most minor intoxications just call for symptomatic care and typically do not necessitate hospitalization. Arrhythmias, considerable chest discomfort, convulsions, extreme agitation, or other symptoms of acute intoxication should all be investigated further in a hospital setting. The unexpected effects and absence of a distinct toxidrome to identify SCs from other recreational drugs make management more difficult due to the lack of an antidote for SCs comparable to that for opioid overdose. Different illnesses, such as hypoglycemia, infections, thyroid hyperactivity, head trauma, and mental disorder, must be ruled out in order to make a differential diagnosis. While the use of haloperidol has also been discussed, caution is suggested in cases of nonspecific agitation. Benzodiazepines are typically sufficient to reduce agitation. Failure with benzodiazepines should provoke thought about final airway control. The main objectives are safeguarding the airway, avoiding rhabdomyolysis, and monitoring for either cardiac or cerebral ischemia, in addition to administering intravenous fluids for dehydration [102,103].

### Limitations

Our study has some limitations, such as incorporating multiple concomitant substances frequently reported in studies, which can generate complex toxicology analysis. Additionally, because toxicology tests and collection sites were not properly monitored, it was impossible to determine fatal doses or harmful effects accurately. We also highlight the lack of clinical studies in the SCs literature, particularly in relation to newer generations of SCs, such as ADB-FUBINACA, AB-PINACA, AB-CHIMINACA, MDMB-CHMICA, and XLR-11. Furthermore, only works in English were considered; as a result, we may have omitted relevant studies written in different languages. Many hazardous effects of SCs go unnoticed since, in many cases, specific toxicological analyses or more active searches are not performed to identify the SCs.

## 5. Conclusions

The deaths and toxic effects linked to SCs have caused concern worldwide due to the severity and duration of their psychological effects, which are still poorly understood and documented in the literature. As a result of their stronger ligation to CB1 receptors than THC, SCs exhibit higher toxicity than THC and have longer-lasting effects. Because of this, their use may be harmful, especially in people with epilepsy and schizophrenia, because of the increased risk of the precipitation of psychiatric and neurologic disorders. In comparison to other drugs, SCs have a high potential to trigger convulsive crisis, a decline in consciousness, and hemodynamic changes. In addition, intoxication by SCs may be challenging to diagnose in EDs due to the clinical syndrome’s similarity to other substances, particularly synthetic opioids, as well as the fact that new SCs emerge annually, making early testing and diagnosis difficult. As a result, it is imperative that more research in this field is conducted in order to clarify the potential harms associated with these substances, warn users of their toxicities and associated risks, particularly those related to NPS, and emphasize the significance of toxicological tests available in EDs that aid in early detection.

## Figures and Tables

**Figure 1 brainsci-13-00990-f001:**
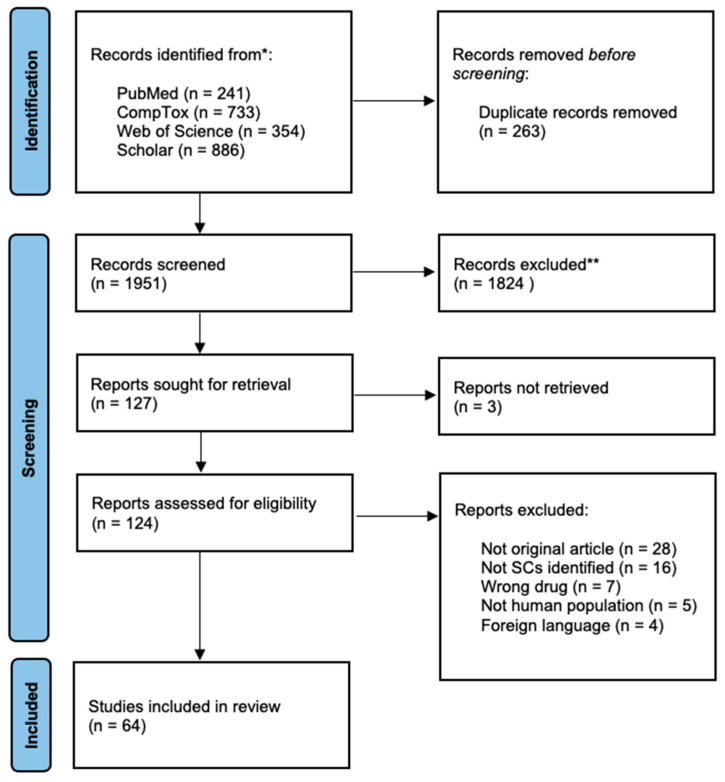
Preferred Reporting Items for Systematic Reviews and Meta-Analyses (PRISMA) flow diagram. * Total: 2214 records. ** Subject excluded (*n* = 996), not human (*n* = 393), other drugs (*n* = 278), in vitro (*n* = 157).

**Figure 2 brainsci-13-00990-f002:**
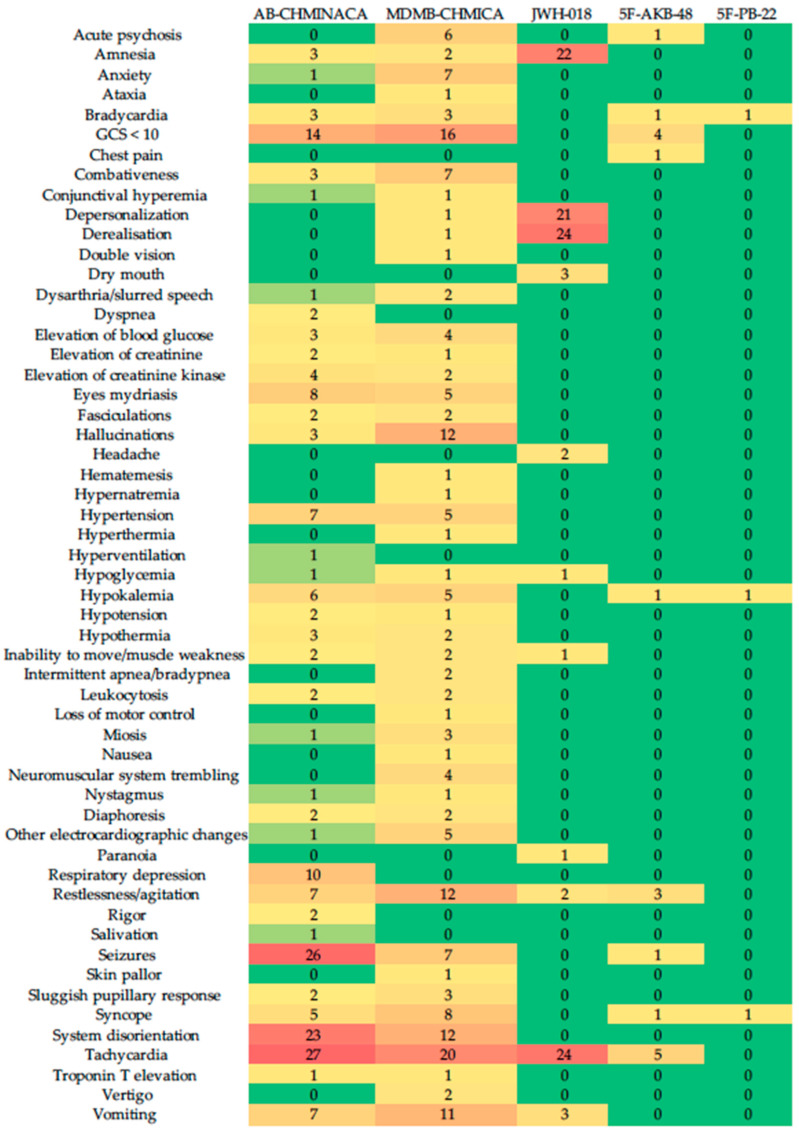
Heatmap of the SCs’ toxic effects in clinical trials.

**Table 1 brainsci-13-00990-t001:** SCs toxic effects in clinical studies.

Author, Year, Country	Synthetic Cannabinoid	N (Sample); Age (Mean); Gender (M:F); Setting (O: Outpatient; I: Inpatient; ED: Emergency Department)	Adverse Effects
Tyndall et al., 2015, EUA [24]	AB-CHMINACA	35; 36; 31:4; ED	Acute delirium and seizures, required ventilator support
Seywright et al., 2016, EUA [25]	MDMB-CHMICA	11; 26; 82% males; ED	Hypothermia, hypoglycemia, syncope, recurrent vomiting, altered mental state, serotonin toxicity
Abouchedid et al., 2017, UK [26]	5F-AKB-48, MDMB-CHMICA, AB-CHMINACA, Cumyl 5F-PINACA, BB-22	18; 31; 79.9% males; ED	Seizures, agitation
Hermanns-Clausen et al., 2017, Germany [27]	AB-CHMINACA	44; 19; 25:4; I	Depression, disorientation, seizures, combativeness, extreme agitation
Kamijo et al., 2018, Japan [28]	AB-CHMINACA	5; 30; NR; O	Tachypnea, tachycardia, hypertension, disturbance of consciousness, agitation, irritability, anxiety, fear, confusion, seizures, psychosis (hallucination, delusion), abnormal behavior
Theunissen et al., 2018, Netherlands [29]	JWH-018	7; 23; 2:4; I	Well tolerated, no serious side effects reported.
Tebo et al., 2019, EUA [30]	AB-FUBINACA, ADB-FUBINACA, AB-CHMINACA, ADB-CHMINACA, 5-flouro-PB-22	132; >18; NR; ED	Nausea, vomiting, anxiety, paranoia, palpitations, seizures, psychosis
Theunissen et al., 2022, Netherlands [31,32]	JWH-018	24; 22; 10:14; I	Internal and external perception, dissociative effects (amnesia, derealization, depersonalization), induced feelings of confusion, impaired psychomotor, divided attention, impulse control
Manini et al., 2022, EUA [33]	5F-MDMB-PICA, ADB-FUBINACA, AB-CHMINACA, AB-FUBINACA, AB-PINACA, MDMB-4en-PINACA, 4F-MDMB-BINACA	29; >18; NR; ED	Acute respiratory failure

NR: not reported.

**Table 2 brainsci-13-00990-t002:** SCs toxic effects in case reports.

Author, Year, Country	Synthetic Cannabinoid	N (Sample); Age (Mean); Gender (M:F); Setting (O: Outpatient; I: Inpatient; ED: Emergency Department)	Adverse Effects
Lapoint et al., 2011, USA [34]	JWH-018	1; 48; 1:0; ED	Seizures, supraventricular tachycardia
Schneir et al., 2011, USA [35]	JWH-018, JWH-073	2; 21; 0:2; ED	Anxiety, palpitation, tachycardia
Young et al., 2012, USA [36]	JWH-018, JWH-073	1; 17; 1:0; ED	Chest pain, tachycardia, bradycardia
Pant et al., 2012, USA [37]	JWH-018	1; 48; 1:0; ED	Seizures, GCS < 10
Freeman et al., 2013, USA [38]	JWH-018	2; 22; 1:1; ED	Ischemic stroke
Hopkins et al., 2013, USA [39]	JWH-018, JWH-073, JWH-122, AM-2201, AM-694	1; 30; 1:0; ED	Cannabinoid hyperemesis syndrome
Patton et al., 2013, USA [40]	AM-2201	1; 23; 1:0; O	Death
Wikström et al., 2013, Sweden [41]	AM-694, AM-2201, JWH-018	1; 26; 1:0; O	Pulmonary edema, death
Chan et al., 2013, UK [42]	6-APB	1; 21; 1:0; ED	Acute psychosis, agitation, paranoid behavior
McQuade et al., 2013, UK [43]	AM-2201	1; 20; 1:0; ED	Seizures
Lemos et al., 2014, USA [44]	XLR-11	1; 22; 1:0; ED	Low body temperature, rigid muscle tone, normal pulse, lack of horizontal and vertical gaze nystagmus, nonconvergence of the eyes, dilated pupil size
Takematsu et al., 2014, USA [45]	XLR-11	1; 33; 1:0; ED	Acute cerebral infarction, hemiparesis, dysarthria, aphasia
Musshoff et al., 2014, Germany [46]	AM-2201, JWH-018, JWH-019, JWH-122, JWH-210, JWH-307, MAM-2201, JWH-122, UR-144	8; 18; 6:2; O	Retarded sequence of movements, lazy, cumbersome, confusion, disorientation, slurred and babbling speech, inappropriate freezing, reduced breathing, enlarged pupils
Gugelmann et al., 2014, USA [47]	PB-22, UR-144	1; 22; 1:0; ED	Seizures
Behonick et al., 2014, USA [48]	5F-PB-22	4; 20; 4:0; O	Pulmonary edema, death
Lonati et al., 2014, Italy [49]	AM-2201	1; 20; 1:0; ED	Excitatory behavior, xerostomia, chest pain, severe dyspnea, tachycardia
Celofiga et al., 2014, Slovenia [50]	AM-2201	4; 28; 4:0; I	Psychosis, anxiety
Shanks et al., 2015, USA [51]	XLR-11	2; 30; 0:2; ED	Anxiety, agitation, hallucinations, hypertension, irritability, seizures, tachycardia
Schwartz et al., 2015 [52]	AB-PINACA	7; 25; 4:3; O	Anxiety, delirium, psychosis, aggressive behaviors, seizures
Hess et al., 2015, Germany [53]	AB-CHMINACA, AB-FUBINACA, AM-2201, 5F-AMB, 5F-APINACA, EAM-2201, JWH-018, JWH-122, MAM-2201	1; 25; 1:0; O	Tachycardia, sedation, psychosis, anxiety and panic attacks, agitation, convulsions, nausea and emesis
Peterson et al., 2015 [54]	AB-CHMINACA, AB-PINACA	58; 28; 55:3; O	Bloodshot eyes (80%), watery eyes (55%), droopy eyelids (68%), horizontal gaze nystagmus (HGN) observed in 50 and 60%, tachycardia
Thornton et al., 2015, USA [55]	AB-PINACA	1; 10; 0:1; ED	GCS < 10
Buser et al., 2016, USA [56]	XLR-11	9; 18; 9:0; I	Nausea and flank or abdominal pain, included two sets of siblings
Adamowicz et al., 2016, Poland [57]	AB-CHMINACA	4; 17; 2:2; ED	Vomiting, seizures, limb twisting, muscle tremors, aggression, agitation, slurred speech, blood pressure spikes, wheezing, respiratory failure, losses of consciousness
Klavž et al., 2016, Slovenia [58]	AB-CHMINACA, AB-FUBINACA	1; 38; 1:0; ED	Dilated pupils, sinus tachycardia, dehydration
Shanks et al., 2016, USA [59]	ADB-FUBINACA	1; 41; 0:1; O	Coronary arterial thrombosis, pulmonary edema, vascular congestion
Barceló et al., 2016, Spain [60]	5F-ADB, MMB-2201	5; 17; 1:4; ED	Psychomotor agitation, confusion, anxiety and psychosis, tachycardia, temporary amnesia, loss of consciousness
Adamowicz, 2016, Poland [61]	MDMB-CHMICA	1; 25; 1:0; ED	Loss of consciousness, asystole
Abouchedid et al., 2016, UK [62]	5F-AKB-48, 5F-PB-22	1; 19; 0:1; ED	Seizures, agitation, tachycardia
Westin et al., 2016, Norway [63]	MDMB-CHMICA	1; 22; 1:0; ED	Arrythmia
Rojek et al., 2017, Poland [64]	AM-2201	1; 18; 1:0; O	Hetero-aggressive behaviour
Angerer et al., 2017, Germany [65]	5F-PB-22, AB-CHMINACA, 5F-ADB	3; 31; 3:0; O	Death
Bäckberg et al., 2017, Sweden [66]	MDMB-CHMICA	9; 34; 8;1; I	Seizures, deep unconsciousness
Meyyappan et al., 2017, UK [67]	MDMB-CHMICA	3; 29; 3:0; ED	Hypercapnia, reduced level of consciousness, seizures, bradycardia
Lam et al., 2017, China [68]	AB-FUBINACA, ADB-FUBINACA	1; 24; 1:0; ED	Somnolence, confusion, agitation, palpitation and vomiting, supraventricular tachycardia
Coppola et al., 2017, Italy [69]	JWH-122	1; 18; 1:0; ED	Hallucinogen persisting perception disorder
Moeller et al., 2017, Germany [70]	ADB-FUBINACA	1; 25; 1:0; ED	Ischemic stroke
Langford et al., 2018, UK [71]	5F-PB-22, 5F-AKB-48	1; 35; 1:0; O	Congestion of the lungs, death
Nacca et al., 2018, USA [72]	ADB-FUBINACA	1; 38; 1:0; I	Encephalopathy, seizure activity, second-degree atrioventricular block type I, respiratory failure, hypotension, hypothermia, hypoglycemia
Maeda et al., 2018, Japan [73]	AB-CHMINACA	1; 29; 1:0; O	Hypoxic encephalopathy and systemic hypoxemia, pulmonary edema, seizures
Yamagishi et al., 2018, Japan [74]	EAM-2201, AB-PINACA	1; 1:0; O	Death
Brandehoff et al., 2018, USA [75]	ADB-FUBINACA	8; 28; 4:4; O	Agitation, delirium, chest pain
Paul et al., 2018, Canada [76]	AB-CHMINACA, UR-144, XLR-11, JWH-022.	2; 15; 2:0; O	Paranoid, dilated cardiomyopathy, cardiomegaly, bilateral pulmonary edema, bilateral pleural effusion, ascites, death
Usui et al., 2018, Japan [77]	MDMB-PINACA	4; 35; 4:0; O	Death
Gaunitz et al., 2018, Germany [78]	MDMB-CHMICA	1; 27; 1:0; ED	Death
Ivanov et al., 2019, Bulgaria [79]	5F-ADB, FUB-AMB	1; 18; 1:0; ED	Acute respiratory distress syndrome (ARDS)
Al Fawaz et al., 2019, Canada [80]	UR-144	1; 19; 0:1; ED	Seizure, cardiomyopathy, respiratory fatigue
Kraemer et al., 2019, USA [81]	5F-ADB	5; 33; 3:2; O	Confusion, psychosis, collapse, loss of consciousness, unsafe driving style, changing moods
Chan et al., 2019, Singapore [82]	ADB-FUBINACA	4; 30; 4:0; ED	Death
Kovács et al., 2019, Hungary [83]	ADB-FUBINACA	1; 23; 1:0; ED	Hypertrophic and dilated heart, severe atherosclerosis of the valves, coronaries and the arteries, edema of the internal organs, death
Hvozdovich et al., 2020, USA [84]	5F-ADB, FUB-AMB, 5F-AMB, MDMB-FUBINACA, AB-CHMINACA	57; 45; 57:0; O	Death
Salle et al., 2021, France [85]	5F-ADB	1; 16; 1:0; ED	Seizure
Simon et al., 2022, Hungary [86]	ADB-FUBINACA	1; 41; 1:0; ED	Myocardial ischemia
Gaunitz et al., 2022, Germany [87]	5F-ADB	1; 31; 1:0; ED	Panic attack

## Data Availability

All supporting data are presented in the manuscript.

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
