# Peer review of "Toxicity of Synthetic Cannabinoids in K2/Spice: A Systematic Review"

_brainsci, 2023, doi:10.3390/brainsci13070990_

Round 1
Reviewer 1 Report
Comments and Suggestions for Authors
The manuscript is well organized, results are clearly presented and references are up-to-date.
There are only some minor issues that should be addressed by authors.
Specific comments and suggestions are given below.
Introduction: Please add number of detected SCs till now and the year of the first appearance on the drug market. Also, highlight that they have totally different chemical structure than THC and that THC is a partial agonist at the CB receptors and many of the SCs are full agonists.
I suggest to use acronym NPS instead NSP for the new psychoactive substances.
Lines 69-81: Please add the most common SC for each generation.
Figure 1: Please explain ‘*’ and ‘**’.
Table 1: I suggest to add range of measured concentration of SC in biological samples if data exist.
Line 135: Please rephrase: ‘’Analyses of SC in blood, serum, or urine samples were performed using electrospray’’ instead of ‘’The toxicologic tests used were Electrospray…’’
Line 166: Is there any reported relation between measured concentration in blood and degree of intoxication?
Line 233: Please explain the acronym SPAs.
In Discussion, please add a few sentences on hospital treatment following intoxication.
Author Response
Dear Reviewer,
We thank the reviewer for such a careful evaluation of our manuscript. We believe your comments have strengthened the paper.
Please find below a point-by-point response to the comments. All the modifications are highlighted in red in the present manuscript.
Reviewer #1
Comment:
“The manuscript is well organized, results are clearly presented and references are up-to-date.
There are only some minor issues that should be addressed by authors.
Specific comments and suggestions are given below.
Introduction: Please add number of detected SCs till now and the year of the first appearance on the drug market. Also, highlight that they have totally different chemical structure than THC and that THC is a partial agonist at the CB receptors and many of the SCs are full agonists.
I suggest to use acronym NPS instead NSP for the new psychoactive substances.
Lines 69-81: Please add the most common SC for each generation.
Figure 1: Please explain ‘*’ and ‘**’.
Table 1: I suggest to add range of measured concentration of SC in biological samples if data exist.
Line 135: Please rephrase: ‘’Analyses of SC in blood, serum, or urine samples were performed using electrospray’’ instead of ‘’The toxicologic tests used were Electrospray…’’
Line 166: Is there any reported relation between measured concentration in blood and degree of intoxication?
Line 233: Please explain the acronym SPAs.
In Discussion, please add a few sentences on hospital treatment following intoxication.”
Response
1) Introduction: Please add number of detected SCs till now and the year of the first appearance on the drug market. Also, highlight that they have totally different chemical structure than THC and that THC is a partial agonist at the CB receptors and many of the SCs are full agonists.
We added the data in the introduction as your suggestion.
“Currently there are hundreds of SCs known and SCs market is constantly evolving and new compounds are continually being developed”
“SCs have greater toxicity due to the binding power of SCs in cannabinoid receptors increasing the chance of side effects. In the early 2000s, substances (e.g., JWH-018) which were primarily created for research purposes started appearing in smoking mixes and these concoctions for smoking quickly gained popularity, particularly in nations where cannabis use for recreational purposes was prohibited or where users wanted to avoid being detected through standard drug testing For this reason, a number of SCs began to be created in secret labs, combined with dried herbal mixes, and sold online as acceptable substitutes for cannabis (or "legal highs"). Known as "K2" (in North America), "Spice" (in Europe), "Youcatan", "Chill" or "Black Mamba" and reportedly safe for eating, these concoctions have been widely marketed as smokable herbal combinations [11].
Since then, SCs’ use has grown considerably which raises concerns about your potential harm. Although often compared to THC, synthetic cannabinoids are not structurally related to the naturally cannabinoids presented in marijuana. Based on earlier SCs structures, four substructures with an indole, indazole, or carbazol core encircled by various N-substituents were used to create SCs. They form a heterogeneous group, most SCs are lipid soluble and non-polar composed of 22 to 26 carbon atoms and very volatile when heated [6].”
2) I suggest to use acronym NPS instead NSP for the new psychoactive substances.
We changed NSP to NPS.
3) Figure 1: Please explain ‘*’ and ‘**’.
* Total: 2214 records
** Subject excluded (n = 996), not human (n = 393), other drugs (n = 278), in vitro (n = 157)
4) Table 1: I suggest to add range of measured concentration of SC in biological samples if data exist.
It was not possible to add because there is a lack of data.
5) Line 135: Please rephrase: ‘’Analyses of SC in blood, serum, or urine samples were performed using electrospray’’ instead of ‘’The toxicologic tests used were Electrospray…’’
We rephrased as suggested.
6) Line 166: Is there any reported relation between measured concentration in blood and degree of intoxication?
There is no report, our aim is to contribute to future investigations.
7) Line 233: Please explain the acronym SPAs.
We corrected to NPS.
8) In Discussion, please add a few sentences on hospital treatment following intoxication.”
We added as below:
“Most minor intoxications just call for symptomatic care and typically do not necessitate hospitalization. Arrhythmias, considerable chest discomfort, convulsions, extreme agitation, or other symptoms of acute intoxication should all be investigated further in a hospital setting. The unexpected effects and absence of a distinct toxidrome to identify SCs from other recreational drugs make management more difficult due to the lack of an antidote for SCs comparable to that for opioid overdose. Different illnesses, such as hypoglycemia, CNS infections, thyroid hyperactivity, head trauma, and mental disorder, must be ruled out in order to make a differential diagnosis. While the use of haloperidol has also been discussed, caution is suggested in cases of nonspecific agitation. Benzodiazepines are typically sufficient to reduce agitation. Failure with benzodiazepines should provoke thought about final airway control. The main objectives are safeguarding the airway, avoiding rhabdomyolysis, and keeping an eye out for either cardiac or cerebral ischemia, in addition to intravenous fluids for dehydration [102, 103].”
Reviewer 2 Report
Comments and Suggestions for Authors
The present article written by Mariana Campello de Oliveira and collegues presents a systematic review of the synthetic cannabinoids in K2/Spice and their toxicology. The present article is interesting, however, it needs much more polishing, in order that the reader can better understand and use the information that is presented.
Some suggestions:
- Extensive English editing
- The introduction should better highlight the background of synthetic cannabinoids, their mechanism of action and implications. Maybe a scheme/table would be helpfull for the reader.
- Methods: Figure 1: "records excluded" Please add more info.
- The Results part is very difficult to read and to be understood. The same applies for the tables and figures. Maybe some of them could be moved to Appendix.
Comments on the Quality of English LanguageExtensive English editing
Author Response
Dear Reviewer,
We thank the reviewer for such a careful evaluation of our manuscript. We believe your comments have strengthened the paper.
Please find below a point-by-point response to the comments. All the modifications are highlighted in red in the present manuscript.
Reviewer #2
“The present article written by Mariana Campello de Oliveira and collegues presents a systematic review of the synthetic cannabinoids in K2/Spice and their toxicology. The present article is interesting, however, it needs much more polishing, in order that the reader can better understand and use the information that is presented.
Some suggestions:
- Extensive English editing
- The introduction should better highlight the background of synthetic cannabinoids, their mechanism of action and implications. Maybe a scheme/table would be helpfull for the reader.
- Methods: Figure 1: "records excluded" Please add more info.
- The Results part is very difficult to read and to be understood. The same applies for the tables and figures. Maybe some of them could be moved to Appendix.”
Response:
1) Extensive English editing
We agree with the reviewer and we made the necessary corrections.
2) The introduction should better highlight the background of synthetic cannabinoids, their mechanism of action and implications. Maybe a scheme/table would be helpfull for the reader.
We have expanded the description of the synthetic cannabinoids, but we were not able to add a table because this would require extensive representation of the four synthetic cannabinoids’ generations.
3) The Results part is very difficult to read and to be understood. The same applies for the tables and figures. Maybe some of them could be moved to Appendix.”
The initial idea of plotting Figure 3 was to visualize the relationship between synthetic cannabinoids and toxicity symptoms. However, the figure showing the results referring to the clinical cases was confusing because it was showing how many papers the symptoms were described in, different from the figure 2 of the clinical reports that described how many patients had the symptoms. In the light of the reviewer’s comments and our choice of outcome, we decided to delete such figure 3 from the present manuscript. We have rewritten the last paragraph of the results in a clearer and more concise manner either.
4) Methods: Figure 1: "records excluded" Please add more info.
We include the reasons for exclusion.
** Subject excluded (n = 996), not human (n = 393), other drugs (n = 278), in vitro (n = 157)
Reviewer 3 Report
Comments and Suggestions for Authors
Brainsci 2430490
Present systematic review entitled “Toxicity of synthetic cannabinoids in K2/Spice: a systematic review” by Mariana Campello de Oliveira et al described the toxicity aspect of synthetic cannabinoids. Instead a systematic review, this looks like a research article (the way it is written and presented). So, it is not sure whether it is review article or research paper. Key words are unnecessarily extra. On certain places choice of word is not clear. If I am correct, Toxic effects in Table 1 should be changed to adverse effect. Inconsistency in table 1 subheading either keep bold or no bold comparable to table 2 (which is nicely presented). Once abbreviation has been used then not need to use full word keep abbreviation.
Line 138 authors have written “central nervous system depression” I am not sure what does it mean.
Concept behind this article is excellent however presentation is not great. Often difficult to follow what author want to convey.
Authors are advised to read and write this article little carefully, they have beautiful content to present.
Comments on the Quality of English LanguageNot bad but need some attention
Author Response
Reviewer #3
Comment:
“Present systematic review entitled “Toxicity of synthetic cannabinoids in K2/Spice: a systematic review” by Mariana Campello de Oliveira et al described the toxicity aspect of synthetic cannabinoids. Instead a systematic review, this looks like a research article (the way it is written and presented). So, it is not sure whether it is review article or research paper. Key words are unnecessarily extra. On certain places choice of word is not clear. If I am correct, Toxic effects in Table 1 should be changed to adverse effect. Inconsistency in table 1 subheading either keep bold or no bold comparable to table 2 (which is nicely presented). Once abbreviation has been used then not need to use full word keep abbreviation.
Line 138 authors have written “central nervous system depression” I am not sure what does it mean. We altered to Glasgow Coma Scale less than 10.
Concept behind this article is excellent however presentation is not great. Often difficult to follow what author want to convey.
Authors are advised to read and write this article little carefully, they have beautiful content to present.”
Response:
Dear Reviewer,
We thank the reviewer for such a careful evaluation of our manuscript. We believe your comments have strengthened the paper.
Please find below a point-by-point response to the comments. All the modifications are highlighted in red in the present manuscript.
Writing about synthetic cannabinoids is challenging because there is a lack of published studies, maybe that is why it looks like a research article, but we worked extensively to make this systematic review, so we hope it has been improved after necessary corrections.
We excluded some of the key words that was unnecessarily extra.
We changed Toxic effects in Table 1 to adverse effects as you recommend.
We altered “central nervous system depression” to Glasgow Coma Scale less than 10.
Round 2
Reviewer 2 Report
Comments and Suggestions for Authors
The authors have addressed all my suggestions.
Reviewer 3 Report
Comments and Suggestions for Authors
No comments , I am happy with author's response.